# Preclinical Large Animal Porcine Models for Cardiac Regeneration and Its Clinical Translation: Role of hiPSC-Derived Cardiomyocytes

**DOI:** 10.3390/cells12071090

**Published:** 2023-04-05

**Authors:** Divya Sridharan, Nooruddin Pracha, Schaza Javed Rana, Salmman Ahmed, Anam J. Dewani, Syed Baseeruddin Alvi, Muhamad Mergaye, Uzair Ahmed, Mahmood Khan

**Affiliations:** 1Department of Emergency Medicine, The Ohio State University, Columbus, OH 43210, USA; 2Department of Internal Medicine, Northeast Georgia Medical Center, Gainesville, GA 30501, USA; 3Lake Erie College of Osteopathic Medicine (LECOM), Erie, PA 16509, USA; 4Department of Chemistry & Biochemistry, The University of Toledo, Toledo, OH 43606, USA; 5Department of Physiology and Cell Biology, The Ohio State University, Columbus, OH 43210, USA

**Keywords:** myocardial infarction, large animal model, porcine, hiPSC-CMs, cell transplantation

## Abstract

Myocardial Infarction (MI) occurs due to a blockage in the coronary artery resulting in ischemia and necrosis of cardiomyocytes in the left ventricular heart muscle. The dying cardiac tissue is replaced with fibrous scar tissue, causing a decrease in myocardial contractility and thus affecting the functional capacity of the myocardium. Treatments, such as stent placements, cardiac bypasses, or transplants are beneficial but with many limitations, and may decrease the overall life expectancy due to related complications. In recent years, with the advent of human induced pluripotent stem cells (hiPSCs), newer avenues using cell-based approaches for the treatment of MI have emerged as a potential for cardiac regeneration. While hiPSCs and their derived differentiated cells are promising candidates, their translatability for clinical applications has been hindered due to poor preclinical reproducibility. Various preclinical animal models for MI, ranging from mice to non-human primates, have been adopted in cardiovascular research to mimic MI in humans. Therefore, a comprehensive literature review was essential to elucidate the factors affecting the reproducibility and translatability of large animal models. In this review article, we have discussed different animal models available for studying stem-cell transplantation in cardiovascular applications, mainly focusing on the highly translatable porcine MI model.

## 1. Introduction

Cardiovascular diseases (CVD), especially myocardial infarction (MI), are the leading cause of mortality in the US [1], irrespective of age, sex, or ethnicity [2]. The World Health Organization has recognized modifiable risk factors, such as sedentary lifestyle, tobacco, and unhealthy dietary practices, paired with psychological stress, as significant contributors to CVDs [3]. MI occurs due to coronary artery constriction, which decreases blood and oxygen availability to the heart muscle, leading to the death of cardiomyocytes [4,5]. Additionally, other reversible ischemic injuries, such as myocardial stunning and hibernation, can also cause reversible ischemia in the cardiac tissue [6]. The ischemic tissue is replaced by a non-contractile scar and remodeling, which compromises cardiac contractility and eventually leads to heart failure [7,8]. While pharmacotherapy may increase survival by minimizing the cardiac remodeling process or providing palliative relief, they do not aid in the regeneration of the dead tissue [9]. Percutaneous Coronary Intervention (PCI) and Coronary Artery Bypass Grafting (CABG) are the approaches commonly adapted to restore perfusion to the injured myocardium [10]. In patients with severe MI, heart transplantation is the only available option. However, the cardiac transplant is limited by donor availability and is further complicated by high rates of immune rejection [11]. New-age therapeutic strategies for CVDs include gene therapy, targeted drug delivery, and stem cell therapies [12,13,14]. While these therapies aim at regenerating the cardiac tissue, most of these strategies are still in their infancy.

Different cell types have been studied for myocardial regeneration by cell-based therapies: multipotent (or adult) stem cells including skeletal myoblasts, hematopoietic stem cells (HSCs), endothelial progenitor cells (EPCs), mesenchymal stem cells (MSCs)and cardiac stem cells (CSCs), human embryonic stem cells (hESCs) and human induced pluripotent stem cells (hiPSCs) [15,16,17]. Among adult stem cells, skeletal myoblasts and MSCs have been studied in numerous preclinical and clinical studies. While these cells showed promising outcomes in improving cardiac function post-transplantation in preclinical studies, the benefits associated with these cells are mainly paracrine in nature [18,19,20]. Additionally, the promising role of these adult stem cells in preclinical studies could not be successfully translated to clinical outcomes, mainly because of poor survival, engraftment, and electrical coupling [21,22,23,24,25,26,27]. In addition to this, most of these adult stem cells are limited by their biological availability, poor in vitro expandability, and immunogenicity [28,29].

On the other hand, hiPSCs are emerging as promising candidates for regenerative medicine applications [28,29,30,31,32]. Both hESCs and hiPSCs can differentiate into any somatic cell type, including neurons, pancreatic cells, hepatocytes, and functional cardiomyocytes [33], and therefore can be used in cell-based therapies, especially for organs, such as the heart, which lack regenerative capabilities [34,35,36,37]. Additionally, autologous stem cell therapy is made possible by the availability of patient-derived hiPSCs, thereby overriding the problems associated with immune rejection [35,38]. While the therapeutic potential of cardiomyocytes differentiated from hESCs (hESC-CMs) or hiPSCs (hiPSC-CMs) have been extensively studied [39,40,41,42,43,44,45,46,47,48,49,50,51,52], their translation into the clinic has been impeded by different factors [53]. For the hiPSC-CMs to be used for cell therapy in patients with MI, different stages need to be optimized (Figure 1). In the last two decades, reproducible protocols for the expansion of hiPSCs [54,55,56] and in vitro differentiation to functional cardiomyocytes (hiPSC-CMs) [57,58,59], and the enrichment of the hiPSC-CMs (≥95% purity) [52,60,61,62] have been developed. The last decade has also shown a rising trend in the use of the hiPSC-CMs in vitro for ‘clinical trials in a dish’ (Figure 1).

Despite this progress, the translation of hiPSCs to the clinic has been somewhat modest. Lack of reproducibility in preclinical studies has been a major impediment to the clinical use of hiPSCs [63]. Our review focuses on translational preclinical and clinical studies using the hiPSC-CMs for myocardial regeneration after MI. Here, we discuss the existing challenges in the use of hESC-CMs and hiPSC-CMs for cardiac regenerative medicine and the need for reliable and reproducible preclinical animal models. Lastly, we discuss how the large animal porcine model may be the critical link for the successful translation of the hiPSC-CMs from bench to bedside and elaborate on the pros and cons of using this model for preclinical studies [64].

## 2. Preclinical Animal Models of MI for Transplantation of hiPSC-CMs

Animal models play a critical role in the successful translation of in vitro studies using cell lines to clinical applications. The critical barrier to the successful use of the hiPSC-CMs in clinics is the lack of knowledge about their long-term survival, engraftment, integration with the host myocardium, and role in the improvement of cardiac function [39]. Different preclinical animal models have been used to test the use of the hiPSC-CMs for myocardial regeneration, including small animal models (mice, rats, guinea pigs) and large animal models (pigs and non-human primates) [51,65,66,67]. Each of these animal models encompasses its respective benefits and limitations.

Small animal models, such as mice, rats, and guinea pigs, are more practical in terms of wide availability, ease of handling, and cost-effectiveness of pre- and post-procedure care [68]. Additionally, the ease and cost-effectiveness of generating transgenic models in small animals make them appealing for use in preclinical studies [69,70]. However, small animals, such as mice and rats have a high heart rate, more than five times of humans, [68,71] which could potentially mask the arrhythmias occurring post-cell transplantation [1]. Other limitations include excessive heart collaterals, large surface area to body ratio, short life span, and genetic variation from humans in terms of contraction and electrophysiology [68,71]. The human ventricular myocytes predominantly express slower β-myosin heavy chain (β-MHC) in comparison to the murine and rodent myocardium that expresses the fast α-MHC, resulting in different contractile properties [68]. Inaccessible cardiac circulation allows permanent or temporary ligation of the coronary artery as the only method for induction of ischemia, which fails to mimic the pathophysiology of naturally occurring ischemia secondary to atherosclerosis [72,73]. Attempts to induce atherosclerosis in rodent coronary arteries have failed to produce the desired results, with atherosclerosis developing in the aortic artery root with no clinical complications [74,75,76].

On the other hand, large animals, such as rabbits, dogs, pigs, sheep, and non-human primates have the advantage of having similar pathophysiology and electrophysiology as humans, and are therefore clinically more relevant [67,77]. However, these models face the challenge of increased expenses, cost of care, and stringent animal rights [78,79,80]. Additionally, large animal models, such as swine, suffer fatal ventricular arrhythmia resulting in high mortality during and immediately after the procedure, affecting sample size and outcomes [81]. Similarly, excessive fat accumulation in Yorkshire pigs with age complicates long-term follow-up [68,81]. Moreover, the complex human pathophysiology influenced by polypharmacy, other comorbidities, age, gender, and genetic disposition, is rarely reproducible in these animal models [82]. However, despite these obstacles, preclinical studies in cardiac regeneration involving animal models are indispensable and are an essential prerequisite to establishing successful clinical translatability. This is corroborated by the increasing trend in the use of large animal models for hiPSC-CM transplantation post-MI (Figure 2).

## 3. The Pig as a Translational MI Model

Among the different large animal models, the pig model is more desirable for use, mainly due to the close resemblance of its anatomy to that of a human. This is evident from the landmark xenotransplantation study carried out in 2022, wherein the heart from a genetically modified pig was transplanted into a patient [83]. Although the patient died after 49 days of the transplantation, the underlying cause was attributed to a latent porcine viral infection. On the other hand, two studies carried out later at the New York University, showed the establishment of normal circulation in patients that received genetically modified porcine hearts, establishing the similarity in anatomy between porcine and human hearts [83].

In this context, the porcine model has become immensely popular as a model for myocardial cell transplantation studies. The heart-to-body weight ratio (5 g/kg) in these animals and their similarities with the human cardiovascular system, makes (e.g., right dominance, and less coronary collaterals) the porcine model a highly translational preclinical model [68,84,85,86]. In addition to the anatomy and physiology being relevant to clinical scenarios, the genetic makeup of these animals, e.g., predominant expression of β-MHC over α-MHC, is also similar to humans [68]. While the porcine model does have some drawbacks, such as their high susceptibility to ventricular fibrillation, high growth rate, and high cost of maintenance and husbandry, their many advantages along with the lesser ethical and humane issues associated with their use, make them a more commonly used large animal model [84,86]. Various studies have made use of the porcine model to determine the translational aspect of the hiPSC-CMs, unlike studies in the small animal models where reproducibility of the experimental outcomes has been a major roadblock. This inconsistency can be attributed to the numerous variabilities in the experimental design, assessment parameters, animal physiology, time and dose of treatment, and animal care. Here, we have reviewed several plausible causes of experimental variability.

### 3.1. Breed of the Animal

Different breeds of pigs have been used for cardiovascular research, ranging from farm breeds, such as Yorkshire and Landrace, to miniature pigs or mini pigs, such as Göttingen and Sinclair [86]. Although each breed has its advantages and disadvantages, one of the most striking differences arises in the growth rate of farm breeds and minipigs. While the farm breeds grow from 1–2 kg at birth to >100 kg at 4 months of age, the minipigs usually grow from 0.5–1 kg at birth to 7–20 kg (depending on the breed) at 4 months of age [87]. Therefore, a careful selection of the porcine breed becomes essential for the study design. Studies to understand the therapeutic efficiency of hiPSC-CMs have made use of both farm pigs as well as minipigs depending on the duration of the study. Farm pigs have been used for studies to understand the safety, early survival, and engraftment (up to 4 weeks) of the transplanted hiPSC-CMs (Table 1) [42,43,48,49,50]. On the other hand, minipigs have been used to determine the long-term cardiac functional assessment (at 8–12 weeks) post-transplantation of the hiPSC-CMs into the ischemic myocardium (Table 1) [44,45,46].

### 3.2. Myocardial Infarction (MI) Model in Pigs

The surgical procedures used to induce MI in the porcine model have also been shown to significantly affect the experimental outcomes [77]. Both open chest procedures (coronary artery ligation and cryoinjury), as well as close chest surgical methods (catheter-guided coronary artery occlusion), have been used for inducing MI in pigs. However, variations in the duration of occlusion, site of occlusion, and duration of reperfusion, among other parameters, are known to influence the infarct site, size, and ultimately the experimental outcome. For instance, occlusion of the LAD is used frequently to induce MI [88,89]. However, since this method is known to cause ventricular fibrillations and therefore higher mortality rates in pigs, many studies have alternatively occluded the left circumflex artery (LCx) to improve the survival of pigs post-MI. While both these methods have been shown to induce infarcts, the site of infarction in both scenarios is markedly different [90]. Another important parameter influencing the size of the infarct is the duration and site of LAD or LCx occlusion. While most studies for the hiPSC-CM transplantation have used the coronary artery occlusion method, the duration and site of occlusion are variable among these studies. No direct correlation has been found between the site of infarct formation and the engraftment of the hiPSC-CMs, which could theoretically have a significant impact on the study outcomes (Table 1).

### 3.3. Timing, Dose, and Method for Cell Delivery

Small animal studies using the hiPSC-CMs have established modest to significant improvement in cardiac function post-transplantation. The functional improvement resulting from hiPSC-CMs, however, is influenced by (a) co-transplantation of other cell types, such as human MSCs, (b) the number of cells transplanted (dose), (c) site of transplantation (infarct and peri-infarct), and (d) route of delivery (intramyocardial injection, patch-based epicardial delivery and hydrogel-mediated delivery). Similarly, these factors have also been shown to play a crucial role in orchestrating experimental outcomes after the transplantation of the hiPSC-CMs in the porcine model.

In a study by Querdel et al. [48], 5–7 cm fibrinogen/thrombin-derived mesh tissue patches containing 4.3–4.5 × 10^8^ hiPSC-CMs, were transplanted onto the pig hearts one week after the induction of MI. Three weeks after transplantation, the hiPSC-CMs were found penetrating all three layers of the myocardium with evidence of advanced sarcomere protein development. In a similar study by Gao et al. [42], two fibrinogen/thrombin patches (4 cm × 2 cm × 1.25 cm) seeded with 4 × 10^6^ hiPSC-CMs along with 2 × 10^6^ hiPSC-derived endothelial cells (ECs) and 2 × 10^6^ hiPSC-derived smooth muscle cells (SMCs), were transplanted epicardially immediately after the induction of MI. Non-ventricular arrhythmias and ST elevation was observed after the first two weeks. However, at the end of the four-week study, a marked improvement in cardiac function was observed with no added disposition to arrhythmia. The scar size was significantly diminished with reduced apoptotic cells in the peri-infarct area.

A study by Lei et al. [50], tested the efficacy of a combination of hiPSC-derived cells seeded on a patch and insulin-like growth factor-1 (IGF-1), releasing gelatin microspheres. A cocktail of 2 × 10^6^ hiPSC-CMs, 2 × 10^6^ hiPSC-ECs, and 2 × 10^6^ hiPSC-SMCs was either intramyocardially injected or delivered on an IGF-1-releasing microsphere-coated fibrin patch into a farm pig after ischemia reperfusion. While arrhythmias were noted during ischemia induction and reperfusion, none were noted at the four-week follow-up. The pigs transplanted with the patch showed increased expression of Nkx2.5 indicative of protection against oxidative damage. Reduced fibrosis, improved wall stress, higher angiogenesis, increased ventricular function, and recovery of cardiac function were observed in the animals transplanted with the patch [50]. The outcomes of the study highlighted the possible method to improve the clinical outcomes of transplanted hiPSC-CMs a fibrin patch loaded with IGF-1.

Kawamura et al. [44,45] fabricated hiPSC-CM-derived cell sheets, which were transplanted as a single layer or multiple layers, four weeks after the induction of MI in minipigs. While transplanted hiPSC-CMs were detected at eight weeks post-transplantation, there was a remarkable decline in their numbers during long-term follow-ups. In a different study, Kawamura et al. used hiPSC-CM sheets in combination with an omental flap in minipigs. First, seven hiPSC-CM sheets were sutured on the epicardium. Then, the omentum was mobilized and sutured to the pericardium and the minipigs were allowed to heal. Follow-up was done at one month, two months, and three months post-cell transplant with serial cardiac magnetic resonance (CMR) imaging, and left ventricular function (end-diastolic volume and end-systolic volume) was assessed with Ejection Fraction (EF).. Improved EF was seen at the end of the follow-up compared to the control group. The technique of using cells in combination with the omental flap showed superior therapeutic efficacy in terms of angiogenesis, secondary to cytokines released and the anti-inflammatory property of the omentum contributing to enhanced cell engraftment. Similarly, Shi et al. [49] conducted a study on a porcine model with a combinatorial therapy of hiPSC-CMs and gelatin microspheres infused with Tb4. Animals were followed up at four and twelve weeks. The cotreatment of Tb4 with hiPSC showed several benefits: increased vasculogenesis, cell proliferation, protection from oxidative stress, improved cardiac function and decreased scar size with increased wall thickness at the area of ischemia.

In addition to the aforementioned factors, the time of cell delivery post-ischemic injury also becomes critical. In most clinical scenarios, transplantation of hiPSC-CMs immediately after MI is not feasible. Hence, the establishment of a timeline for cell transplantation may largely affect the translatability of the experimental outcomes to clinical implementation. Most studies on transplantation of hiPSC-CMs in the pig model for MI involve delivering the cells one to four weeks after induction of MI [44,45,46,47,48,91,92,93,94] (Table 2), with a few exceptions where the transplantation is performed within an hour of inducing MI in the pigs [42,43,49]. The former provides a more clinically relevant scenario where the cardiac function is sufficiently compromised to test the regenerative potential of the transplanted cells.

### 3.4. Immunosuppression

For the successful survival and engraftment of transplanted allogenic stem cells, overriding the host’s immune system is inevitable. In this regard, a major advantage of the use of small animal models for testing the safety and efficacy of stem cell therapy is the availability of transgenic immunocompromised animals [95]. The availability of these genetically modified animal models minimizes variability in the immunosuppression regime as well as eases the studies for long-term survival, engraftment of transplanted cells, and therefore their associated beneficial functional outcomes. However, unlike small animal models, the severe combined immunodeficiency (SCID) pigs, such as the ARTEMIS^−/−^ pigs, are very expensive and require a very specialized positive pressure room to maintain a higher air pressure than the surrounding environment, filtered air, and water to protect from pathogens [96]. Nevertheless, immunocompromised porcine models mimic the clinical scenario for allogenic cell transplantation studies and therefore, can be used for translation study designs, although they are not economical. On the other hand, several studies have used immunosuppressants, such as cyclosporine or tacrolimus, to suppress the immune system in wild-type pigs (Table 1) [97]. However, the most common method is the use of a proper immunosuppressive drug regime to override the host’s immune response. However, unlike small animal models, the immunosuppression regime in the porcine model has not yet been fully established. A plethora of drugs, dosages and durations have been used in different studies [38,98], which may have a significant impact on the outcomes of the study (Table 1). Additionally, while this is a viable option of preclinical studies, long term use of immunosuppressants for patients may not be desirable due to the plausible detrimental side effects [99,100].

One method to overcome the immunogenicity of hiPSCs is to make use of human leukocyte antigen (HLA)-matched donor cells for transplantation. In recent years, attempts are being made to bank hiPSCs derived from donors with homozygous human leukocyte antigen (HLA) haplotype [38,101]. Another approach involves genetically ablating the HLA molecules in hiPSCs to minimize their immunogenicity underway [38,101]. Furthermore, other approaches modulate the immune responses of the host’s immune system using small molecules or growth factors [50].

## 4. Clinical Trials on hiPSC-CMs as a Therapy for Cardiac Regeneration

Currently, bench research is challenged by the lack of representative disease models [77] and time lag, and is not always translatable into clinical medicine. Factors contributing to the increased incidence of CVD include genetic and epigenetic factors, as well as environmental factors and lifestyle changes, which makes the animal models (small and large) less suitable to be studied for human subjects. With 90% of animal model studies failing to be translatable clinically, bench research on animals seems a less promising option. However, the role of hiPSC-derived cardiac cells has been long studied in different CVDs, e.g., Long QT Syndrome, Hypertrophic Cardiomyopathies, Marfan Syndrome, Dilated Cardiomyopathy, and more [102]. The United States offers the highest contribution of 36% to clinical studies globally, followed by France and China. In interventional studies involving re-transplantation of tissues in the human heart, the US accounts for 16.7% of the cases, with China leading at 36.7% [34]. Currently, four clinical trials are underway using hiPSC-CMs to treat patients with CVDs. NCT04982081 is now a phase I clinical trial that started in August 2021 in Xijing Hospital, China, and is currently recruiting participants for a randomized, double-blinded interventional trial where twenty participants will be subjected to parallel assignment into two groups. One group will receive 100 million hiPSC-CMs injected via a transcatheter endocardial injection system, and the second group will receive 400 million hiPSC-CMs via the same delivery method. The primary endpoint includes incidence of serious adverse events (SAE): death, fatal myocardial infarction, stroke, tamponade, cardiac perforation, ventricular arrhythmias affecting hemodynamics (>15 s), and tumorigenicity related to the hiPS-CM follow-up until the end of first month post-catheterization. Secondary outcomes include seven categories assessed up to twelve months post-catheterization. These categories include incidence of arrhythmia for the first six months; changes in panel reactive antibodies and donor-specific antibodies at one, three, and six months post-catheterization; assessment of left ventricular systolic performance via PET/CT scan at six and twelve months, and via MRI at one, three, six and twelve month;, the incidence of tumor formation; functional status using the six-minute walk test and using the New York Heart Association (NYHA) at one, three, six and twelve months. The study is planned to be completed by December 2023, but no study results have been posted. NCT04945018 is a phase I/II interventional clinical trial that intends to recruit ten participants in an open-label model with a sequential assignment of participants. A hiPSC-CM spheroid suspension will be injected into participants in low and high doses via specialized needles for implantation and guided adaptors. The primary outcome includes monitoring adverse events up to 26 weeks post-transplantation. Secondary outcomes will be measured at 26 and 52 weeks post-transplantation, including left ventricular ejection fraction via cardiac MRI and echocardiography, myocardial wall motion in echocardiography, myocardial blood flow and viability in SPECT, six-minute walk distance, Kansas City Cardiomyopathy Questionnaire (KCCQ), five-level EQ-5D-5L, and N terminal Pro-brain Natriuretic Peptide (NT-proBNP). The study has not started recruiting, but the primary completion date is 30 September 2023. NCT 04396899 is an interventional phase I/II clinical trial being conducted at the University Medical Center, Gottingen. The recruitment is in progress, and the study is currently recruiting 53 participants in a single group assignment with an open-label model. The trial, called BioVAT-HF, will use Engineered Heart Tissue (EHT) made from a defined mixture of hiPSC-CMs and stromal cells in a bovine collagen type I hydrogel. The target patients are those with severe HF and EF <35% and no option of cardiovascular transplant. The primary outcome will target heart wall thickness, and heart wall thickening fraction monitored via high-resolution ECHO or CINE-mode MRI monitored at two weeks, one month, three months, six months, and twelve months after transplantation. The estimated completion date is October 2024. NCT 04696328 is an interventional clinical trial conducted at the Osaka University where ten participants will be recruited in a single group assignment, open-labeled model. Each participant will undergo the transplantation of a hiPSC-CM sheet. The primary outcome is the assessment of left ventricular systolic function (LVEF), and blood tests will assess safety. Secondary outcomes will be left ventricular structural and functional evaluation, NYHA classification evaluation, specific activity scale, Minnesota living with heart failure questionnaire, SF-36, six-minute walk test, BNP, NT-proBNP, and exercise tolerance at 26 and 52 weeks post-transplantation.

## 5. Conclusions and Future Directions

In summary, advancements in technology and contemporary bioengineering tools have advanced the use of hiPSCs in treating damaged myocardium after MI. However, the unavailability of autologous hiPSC lines for each patient or an immune-privileged hiPSC line and the lack of highly reproducible large animal models, limit the clinical translatability of hiPSCs for CVDs. Nonetheless, with the increasing prevalence of CVDs globally, optimization of tools and techniques for the transplantation of hiPSCs is imperative. While the porcine model holds immense potential as a link between bench and bedside, overcoming its many limitations, discussed in this review may pave the way for better clinical translation of hiPSC-CMs.

## Figures and Tables

**Figure 1 cells-12-01090-f001:**
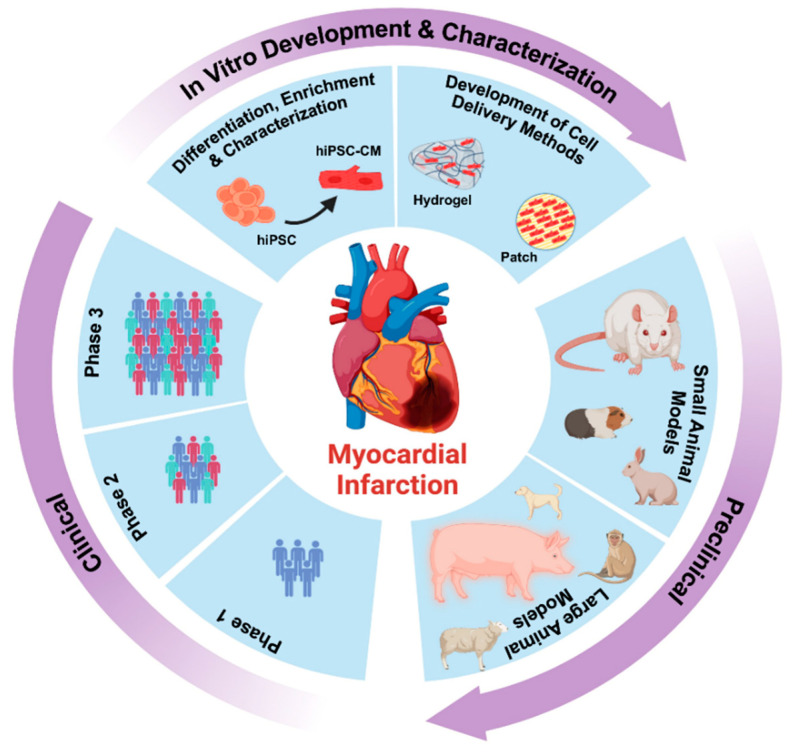
Stages for the translation of hiPSC-CMs from bench to bedside.

**Figure 2 cells-12-01090-f002:**
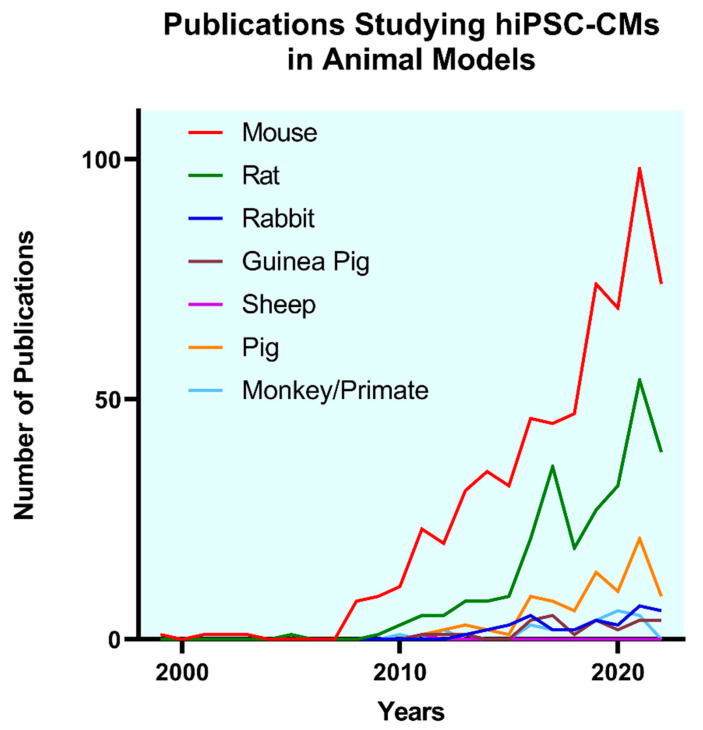
Annual number of publications involving different preclinical small and large animal models for hiPSC-CM transplantation studies (Source: Scopus).

**Table 1 cells-12-01090-t001:** Preclinical studies on iPSCs in small and large animal models.

	Cell Type	Animal Model	Model of MI Induction	Route of Delivery (IM/IV/IC)	No. of Transplanted Cells	Time of Delivery after MI	Immunosuppression (Dose/Time)	Study Duration	Result	Reference
1	hiPSC-CMs	Farm pigs (Wild type or LEA29Y-overexpressing transgenic pigs)	N/A	Fibrin patch (5 × 7 cm)	450 × 10^6^ cells	7 days	Methylprednisolone (250 mg on the day of surgery followed by 125 mg/day, daily, starting on day 1-post surgery) Tacrolimus (0.2 mg/kg bw, daily) Mycophenolate Mofetil (40 mg/kg bw, 2 doses, 1st: day of surgery, 2nd: 4th-day post-surgery) Belatacept (10 mg/kg bw only in wildtype pigs, 2 doses: 1st: day of surgery, 2nd: 4 days post-surgery)	1–2 weeks	Standard pharmacological suppression (wild-type pigs) resulted in poor transplant survival. Better transplant survival was observed in transgenic pigs.	[49]
2	hiPSC-CMs with hiPSC-ECs and hiPSC-SMCs	Yorkshire pigs	Distal LAD artery occlusion for 60 min, followed by reperfusion	Fibrin patch (4 cm × 2 cm × 1 cm)	Two patches per animal, each containing 4 × 10^6^ hiPSC-CMs, 2 × 10^6^ hiPSC-ECs, 2 × 10^6^ hiPSC-SMCs	Immediately after reperfusion	Cyclosporine (15 mg/kg, daily) Methylprednisolone (1.5 mg/kg bw, daily)	4 weeks	Improved LV function, reduced scar size, increased cardioprotection, and enhanced angiogenesis. No ventricular arrhythmias reported	[43]
3	hiPSC-CM, hiPSC- ECs, hiPSC-SMCs	Yorkshire pigs	LAD and LCx occlusion for 60 min followed by reperfusion for 15 min	Intramyocardial injection or IGF-1 loaded fibrin patch	2 × 10^6^ hiPSC-CMs, 2 × 10^6^ hiPSC-ECs, 2 × 10^6^ hiPSC-SMCs	Immediately after reperfusion	Cyclosporine (15 mg/kg bw, daily, starting 3 days before surgery)	4 weeks	Improved LV function. No ventricular arrhythmias Better engraftment with patch transplantation compared to direct cell injection	[51]
4	hiPS-CMs with hMSCs	Mini pigs	Ameroid Constrictor placed around LAD coronary artery	Scaffold- free cell sheets with omentum flap	5 × 10^6^ hMSCs, hiPSC-CMs: NA	4 weeks	Tacrolimus (0.75 mg/kg, daily starting 5 days before transplantation), Mycophenolate Mofetil (500 mg, daily, starting 5 days before transplantation) and Prednisolone (20 mg, daily, starting 5 days before transplantation)	8 weeks	Improved LVEF and cell engraftment	[45,46]
5	hiPSC-CM	Mini-pigs	Ameroid constriction of LAD	Scaffold-free cell sheets	N/A	4 weeks	Tacrolimus (0.6 mg/kg bw, daily, starting 5 days before transplantation)	8 weeks	Improved cardiac function, Signs of increased angiogenesis No Teratoma formation	[47]
6	hiPSC-CMs	Yorkshire-landrace Pigs	Permanent ligation of LAD and LCx arteries	Intramyocardial injection through epicardially placed fibrin/thrombin patch	1.2 × 10^8^ hiPSC-CMs with or without Thymosin β4 (Tb4)-releasing gelatin microspheres	Immediately post-ligation	Cyclosporin (15 mg/kg bw, daily, starting 3 days before surgery)	4 weeks	Improved LV systolic function, higher cell engraftment enhanced angiogenesis and reduced infarct size in pigs that received both hiPSC-CMs and Tb-4-releasing microspheres No ventricular arrythmia or tumor formation	[50]
7	hiPSC-CMs, hiPSC-ECs, hiPSC-SMCs	Pigs	LAD artery occlusion for 60 min followed by reperfusion	Intramyocardial delivery	10 × 10^6^ hiPSC-CMs, 5 × 10^6^ hiPSC-ECs, 5 × 10^6^ hiPSC-SMCs or 7.5 mg exosomes derived from these cells	Immediately after reperfusion	Cyclosporine (15 mg/kg, daily) Methylprednisolone (1.5 mg/kg, daily)	4 weeks	Improved cardiac function, increased angiogenesis, and higher cellular metabolism was observed in the pigs which received cells or exosomes treatment.	[44]
8	hESC-CMs	Farm pigs	LCX obstructed with balloon and 700 microspheres injected to induce ischemia, At 4-week ventricular pacing for an additional 4 weeks to induce HF	Intramyocardial injection	2 × 10^8^ hESC-CMs or 2 × 10^8^ hiPSC-MSC	8 weeks	Oral steroid (40 mg/day, daily, starting 3 days before transplantation), Cyclosporine (200 mg/day, daily, starting 3 days before transplantation)	8 weeks	Improvement in LV function with hiPSC-MSC administration showed more improvement in cardiac function than the hESC-CMs group	[48]

**Table 2 cells-12-01090-t002:** Current clinical trials on hiPSC-CMs in patients with heart failure (Source: https://clinicaltrials.gov, accessed on 9 December 2022).

Title	Phase	Study Type	Status	Identifier	PI
Treating Congestive Heart Failure Patients with hiPSC-CMs Through Catheter-based Endocardial Injection	Phase I	Interventional	Recruiting	NCT04982081	Ling Tao, MD, Ph.D. Xijing Hospital
A Phase I/II Study of hiPSC-CM Spheroids (HS-001) in Patients with Severe Heart Failure, Secondary to Ischemic Heart Disease or LAPis study.	Phase I Phase II	Interventional	Not yet recruiting	NCT04945018	None
Safety and Efficacy of hiPSC derived Engineered Human Myocardium as Biological Ventricular Assist Tissue in Terminal Heart Failure (BioVAT-HF)	Phase I Phase II	Interventional	Recruiting	NCT04396899	Tim Seidler, Prof. University Medical Center Goettingen
Clinical Trial of Human (Allogeneic) hiPSC-CM Sheet for Ischemic Cardiomyopathy	Phase I	Interventional	Recruiting	NCT04696328	Yoshiki Sawa, Ph.D. Osaka University

## Data Availability

Not applicable.

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
