# Peer review of "Preclinical Large Animal Porcine Models for Cardiac Regeneration and Its Clinical Translation: Role of hiPSC-Derived Cardiomyocytes"

_cells, 2023, doi:10.3390/cells12071090_

Round 1

Reviewer 1 Report

This is a nice literature overview on the use of pig models for the study of stem cell transplantation, notably by human iPS, for repair of myocardial infarction.

I have a few suggestions for further improvement.

·        In the Introduction, a bit more detailed information on the underlying myocardial substrate would be good, i.e. hibernation (see Nat Rev Cardiol 7, 2021, 522-36) and remodeling (see Lancet 383, 2014, 1933-43). Please also refer to Nat Rev Cardiol 12, 2020, 773-89 for myocardial infarction in l. 36.

·        For a general overview on the pig model of myocardial ischemia/reperfusion, please refer also to JMCC 6, 2011, 951-63.

·        For translation of cardioprotection and the use of animal models (l. 87/88), please refer also Circ Res 120, 2017, 1477-86.

·        For problems in the use of in vitro human iPS for the study of cardioprotection, see and refer also to Basic Res Cardiol 116, 2021, 34.

·        For the use of cyclosporine, see and refer also to AJP 323, 2022, H904-16.

Author Response

We would like to thank the reviewers for their feedback and comments to enhance the impact of our manuscript. We have addressed their comments in our manuscript (highlighted using track changes) and included detailed responses below.

Reviewer 1

  • In the Introduction, a bit more detailed information on the underlying myocardial substrate would be good, i.e. hibernation (see Nat Rev Cardiol 7, 2021, 522-36) and remodeling (see Lancet 383, 2014, 1933-43). Please also refer to Nat Rev Cardiol 12, 2020, 773-89 for myocardial infarction in l. 36.

We have included these references (Ref No. 5, 6 and 8)

  • For a general overview on the pig model of myocardial ischemia/reperfusion, please refer also to JMCC 6, 2011, 951-63.

We have included this reference (Ref No. 67)

  • For translation of cardioprotection and the use of animal models (l. 87/88), please refer also Circ Res 120, 2017, 1477-86.

We have included this reference (Ref No. 65)

  • For problems in the use of in vitro human iPS for the study of cardioprotection, see and refer also to Basic Res Cardiol 116, 2021, 34.

We have included this reference (Ref No. 64)

  • For the use of cyclosporine, see and refer also to AJP 323, 2022, H904-16.

We have included this reference (Ref No. 98)

Reviewer 2 Report

Overall the review is well written and provides an acceptable up-to-date knowledge regarding the pre-clinical investigation of iPSC-CM and clinical translation. It summarized the possible causes of experimental variables that affect the inconsistency of the animal models. Bellow are some comments to improve the manuscript.

1. Lane 26, "for studying stem transplantation in cardiovascular application" is not clearly described, it needs to be revised.

2. Lane 35, MI already first abbreviated in lane 31. So don't need to provide the full name of MI.

3. lane 49, "multipotent (adult) stem cells" expressed as " multipotent (or adult) stem cells"?

4. lane 97, The next sentence after "but are met with other challenges." seems expressed the pros of small animal model. so it's better to express as "while met with other challenges".

5. lane 177,  The expression of "For. e.g." should be corrected as "For example, "

6. Lane 197, not need to highlight the research lead by " Querdel et al" in bold.  Also lane 201/208/220/224/233 for the rest in the manuscript.

7, Lane 203, hiPSC-ECs and hiPSC-SMCs, should give the full description as it first described in the manuscript.

8, lane 204, "ST elevation". what's the full name for "ST" .

9, Lane 224, reference missing.

10, Table 1, the  the cell numbers should be in the right format.

Author Response

We would like to thank the reviewers for their feedback and comments to enhance the impact of our manuscript. We have addressed their comments in our manuscript (highlighted using track changes) and included detailed responses below.

Reviewer 2

  1. Lane 26, "for studying stem transplantation in cardiovascular application" is not clearly described, it needs to be revised.

We have edited the statement for clarity.

  1. Lane 35, MI already first abbreviated in lane 31. So don't need to provide the full name of MI.

We have incorporated the correction.

  1. lane 49, "multipotent (adult) stem cells" expressed as " multipotent (or adult) stem cells"?

We have incorporated the suggested change.

  1. lane 97, The next sentence after "but are met with other challenges." seems expressed the pros of small animal model. so it's better to express as "while met with other challenges".

We have incorporated the suggested change.

  1. lane 177,  The expression of "For. e.g." should be corrected as "For example, "

We have incorporated the suggested change.

  1. Lane 197, not need to highlight the research lead by " Querdel et al" in bold.  Also lane 201/208/220/224/233 for the rest in the manuscript.

We have incorporated the suggested change.

7, Lane 203, hiPSC-ECs and hiPSC-SMCs, should give the full description as it first described in the manuscript.

     We have incorporated the suggested change.

8, lane 204, "ST elevation". what's the full name for "ST" .

    ST elevation refers to the “ST” segment of the ECG. Therefore, it does not have an expanded terminology.

9, Lane 224, reference missing.

We have inserted the reference.

10, Table 1, the  the cell numbers should be in the right format.

We have formatted the numbers correctly.

Reviewer 3 Report

In the presented review, the authors have reviewed porcine models for cardiac regeneration and the current drawbacks of such models. The manuscript then dives into the current clinical trials on hiPSC-derived cardiomyocytes. 

I have one minor comment- There are two fairly recent reviews in the area

1. Bizy A, Klos M. Optimizing the Use of iPSC-CMs for Cardiac Regeneration in Animal Models. Animals (Basel). 2020 Sep 2;10(9):1561. doi: 10.3390/ani10091561. PMID: 32887495; PMCID: PMC7552322. 

2. Martínez-Falguera D, Iborra-Egea O, Gálvez-Montón C. iPSC Therapy for Myocardial Infarction in Large Animal Models: Land of Hope and Dreams. Biomedicines. 2021 Dec 5;9(12):1836. doi: 10.3390/biomedicines9121836. PMID: 34944652; PMCID: PMC8698445.

I understand the presented manuscript focuses on porcine models in preclinical testing and how the information is translated in clinical trials. However, I feel the authors should cite the above said reviews and what is the need of the current review as compared to them

Author Response

We would like to thank the reviewers for their feedback and comments to enhance the impact of our manuscript. We have addressed their comments in our manuscript (highlighted using track changes) and included detailed responses below.

  1. Bizy A, Klos M. Optimizing the Use of iPSC-CMs for Cardiac Regeneration in Animal Models. Animals (Basel). 2020 Sep 2;10(9):1561. doi: 10.3390/ani10091561. PMID: 32887495; PMCID: PMC7552322. 

We have citied this paper in the text (Ref No. 53).

  1. Martínez-Falguera D, Iborra-Egea O, Gálvez-Montón C. iPSC Therapy for Myocardial Infarction in Large Animal Models: Land of Hope and Dreams. Biomedicines. 2021 Dec 5;9(12):1836. doi: 10.3390/biomedicines9121836. PMID: 34944652; PMCID: PMC8698445.

We have citied this paper in the text (Ref No. 68).